# Collagen-Based Drug Delivery Agents for Glioblastoma Multiforme Treatment

**DOI:** 10.3390/ijms26136513

**Published:** 2025-07-06

**Authors:** Barbara Guzdek, Kaja Fołta, Natalia Staniek, Magdalena Stolarczyk, Katarzyna Krukiewicz

**Affiliations:** 1Department of Physical Chemistry and Technology of Polymers, Silesian University of Technology, M. Strzody 9, 44-100 Gliwice, Poland; bg308082@student.polsl.pl (B.G.); kajafol131@student.polsl.pl (K.F.); ns301155@student.polsl.pl (N.S.); ms301156@student.polsl.pl (M.S.); 2Chemistry Students Research Society, Faculty of Chemistry, Silesian University of Technology, M. Strzody 9, 44-100 Gliwice, Poland; 3Centre for Organic and Nanohybrid Electronics, Silesian University of Technology, Konarskiego 22B, 44-100 Gliwice, Poland

**Keywords:** collagen, drug delivery, glioblastoma, brain tumor, hydrogel, regional chemotherapy

## Abstract

Being one of the most aggressive primary brain tumors, glioblastoma multiforme (GBM) is known from the median survivals of just 15 months following diagnosis. Conventional treatments, including surgical resection, radiotherapy, and chemotherapy, have limited efficiency due to the invasive nature of glioma cells and the presence of a blood–brain barrier. Therefore, adjuvant therapy in the form of a localized delivery of chemotherapeutic agents is indispensable to increase the chances of patients. Among a variety of advanced drug carriers, collagen has recently emerged as an excellent choice for regional chemotherapy, mainly due to its biocompatibility, biodegradability, weak antigenicity, biomimetics, and well-known safety profile, as well as its native presence in the extracellular matrix of the central nervous system. The aim of this paper is to highlight the most recent studies describing the application of collagen as a drug carrier able to provide an extended delivery of chemotherapeutic agents directly to the GBM site, and to provide exciting opportunities for its future applications.

## 1. Introduction

Glioblastoma multiforme (GBM), originating from astrocytic glial cells, is one of the most aggressive primary brain tumors [1]. Due to the non-specificity of the symptoms, which include headaches, ataxia, dizziness, vision, and frequent syncope, it is often misdiagnosed leading to the delay in treatment [2]. Characterized with intra- and intertumor heterogeneity, GBM is resistant to treatment and known for high tumor recurrence rates [3]. Although surgery is currently the first choice for tumor debulking, this treatment alone leads to median survivals of only 3 to 6 months [4]. What is more, tumor recurrence, often occurring within 2 cm of the surgical cavity, is attributed to the invasive nature of glioma cells, which infiltrate surrounding brain tissues and resist complete surgical removal. Therefore, adjuvant therapy is indispensable to increase the chances of patients. In the case of a brain tumor, conventional chemotherapy is not considered as efficient due to the presence of a blood–brain barrier (BBB), which hinders transport of small molecules, including cytotoxic drugs. This challenge can be, however, overcome by the use of novel drug delivery methods, including biopolymers and nanoparticles, that are able to efficiently pass the BBB and provide a controlled supply of therapeutic agents, allowing for a spatially and temporally controlled chemotherapeutic treatment [5].

Among a variety of advanced drug carriers, biopolymers have recently emerged as an excellent choice for regional chemotherapy, mainly due to their low toxicity, non-immunogenic behavior, and tissue and cell compatibility [6]. Collagen, particularly, has been widely explored as a drug carrier, due to its biocompatibility, biodegradability, weak antigenicity, biomimetics, and well-known safety profile [7]. Recent research has shown that collagen can be successfully applied in GBM treatment, mainly because of its native presence in the extracellular matrix of the central nervous system. Additionally, it can be processed using various techniques, resulting in the formation of drug carriers of different sizes and shapes, giving an opportunity for a personalized treatment. The aim of this paper is to highlight the most recent studies describing the application of collagen as a drug carrier able to provide an extended delivery of chemotherapeutic agents directly to the GBM site, and to provide exciting opportunities for its future applications.

## 2. Glioblastoma Multiforme

Glioblastoma multiforme accounts for nearly half of all primary malignant brain tumors, yet its annual incidence remains relatively low, at approximately 3.2 per 100,000 individuals [8]. According to the WHO classification, GBM is categorized as a grade IV tumor, with an average survival time ranging between 6 and 12 months [2]. GBM originates when the oncogenic pathways affect neural stem cells, glial precursor cells, or neural crest (NC)-derived perivascular mesenchymal stromal cells (Figure 1) [9]. GBM is frequently incurable, and the median overall survival for patients is about 15 months [10]. Early symptoms of brain tumors in adults are often nonspecific, leading to multiple consultations in primary care before more specialized diagnostics are pursued. Common symptoms include headaches that intensify in a recumbent position, often accompanied by vomiting, visual disturbances, or cognitive impairments [11].

Magnetic resonance imaging (MRI) remains the diagnostic tool of choice, and in many cases, tumors are identified when they reach a diameter of approximately 4 cm. The standard of care for newly diagnosed neuroblastoma includes tumor resection, radiation therapy, and chemotherapy (Figure 2) [12]. However, poor GBM prognosis is largely due to the tumor’s highly proliferative and infiltrative nature, which prevents complete surgical resection [13]. Radiotherapy is often ineffective, hindered by the BBB and the presence of therapy-resistant cells in hypoxic tumor regions [14]. The standard treatment includes maximal surgical resection followed by radiotherapy and chemotherapy, which together may extend survival up to 202 weeks [15]. Temozolomide, administered alone or in conjunction with radiotherapy, has shown to prolong survival with minimal additional toxicity [16]. Adjuvant therapies include corticosteroids for vasogenic edema and antiepileptic drugs for seizure management [11].

The development of innovative therapeutic approaches is the area that is currently extensively explored. For instance, dendritic cell-based vaccines derived from autologous cells have shown a beneficial impact on short-term survival [17]. RNA interference (RNAi) represents another promising strategy, enabling complete or partial silencing of specific gene expression. When combined with standard treatments, RNAi has improved outcomes by limiting the invasive potential of GBM cells [18,19]. A significant advancement in immunotherapy is the use of chimeric antigen receptor (CAR) T-cell therapy. This involves engineering autologous T cells to express receptors targeting tumor-specific antigens such as IL13Rα2, EGFRvIII, or HER2, and reinfusing them into the patient. While CAR T-cell therapy has yielded remarkable results in hematologic malignancies, early studies in GBM have shown limited efficacy. This may be attributed to the scarcity, heterogeneity, and immune-driven loss of tumor specific antigens. Nevertheless, case studies have reported complete intracranial and spinal tumor regression following CAR T-cell therapy targeting IL13Rα2. These findings suggest potential therapeutic benefits, despite some instances of tumor recurrence due to diminished IL13Rα2 expression [20,21]. Contemporary therapeutic strategies also increasingly employ nanotechnology to overcome the limitations of the BBB. Liposomal nanoparticles, due to their biocompatibility, capacity for drug encapsulation, and prolonged systemic circulation, are under extensive investigation. Functionalization of liposomes with ligands—such as antibodies or receptors overexpressed in GBM cells—has significantly enhanced therapeutic efficacy with minimal increases in systemic toxicity [20].

## 3. Role of Proteins in the Brain

The extracellular matrix (ECM) constitutes a three-dimensional framework that is fundamental to the architecture and function of all human tissues [22]. It is a complex network composed of proteins, proteoglycans, and hyaluronan, secreted by the resident cells (Figure 3) [23]. While it was long regarded primarily as a structural element, it is now well established that the ECM plays a crucial role in regulating cellular functions—including migration, differentiation, survival, and maintenance of tissue homeostasis. Cells possess receptors for ECM components that mediate these processes [24]. Damage to the ECM caused by acute or chronic injury can trigger a cascade of healing events, commonly referred to as the “wound healing cascade.” These overlapping processes begin with the cessation of bleeding and clot formation, known as hemostasis, followed by an inflammatory phase that transitions into a proliferative phase characterized by abundant granulation tissue. Finally, a remodeling phase occurs, during which collagen type I-rich tissue is formed. Each of these stages is closely linked to cellular signaling, in which ECM plays a major role [22].

The ECM has a particular significance in the brain, especially in the basement membranes surrounding blood vessels, where collagen type IV helps stabilize the BBB while supporting the adhesion, migration, and survival of glial and neuronal cells. Although collagens, the most abundant proteins in the human body, are present in the brain ECM in limited amounts, they significantly influence signaling processes such as neurogenesis and the inflammatory response [24]. Due to its properties, including high biocompatibility, biodegradability, non-toxicity, and the ability to form structures with suitable mechanical characteristics, collagen is a particularly attractive biomaterial in neural tissue engineering. It can be processed into gels, sponges, microspheres, nanoparticles, or carriers that support regeneration. Collagen enables the localized delivery of drugs, growth factors, and stem cells, thereby creating an environment conducive to the repair of damaged central nervous system structures. Moreover, the development of recombinant collagen technologies allows for the production of highly pure and reproducible materials, eliminating the risk of pathogen transmission from animal-derived sources [25].

Combining collagen with glycosaminoglycans, major components of the brain ECM, opens new avenues for designing biomaterials that structurally and functionally replicate native neural tissue. Such composites may effectively support regeneration by improving cellular integration and enhancing biological activity [26]. ECM-derived materials and their analogues, such as collagen and hyaluronic acid, also show great promise as carriers for drugs and cells in other areas of regenerative medicine. Their biological compatibility and tunable mechanical properties make them attractive agents for the treatment of skin disorders, cartilage injuries, and heart failure. With their capacity for precise, controlled therapeutic delivery, these materials are becoming a cornerstone of modern therapeutic strategies. Electrospun scaffolds, including those based on collagen, offer increased surface area for efficient drug release, such as chemotherapeutic agents. Collagen used in these systems may contribute to inhibiting tumor progression. Coaxial electrospinning enables drug encapsulation, such as of doxorubicin, opening new possibilities for cancer treatment [27].

## 4. Collagen

### 4.1. Chemistry of Collagen

Collagen is the main component of the skin, especially the dermis of mammals. To date, 28 types of collagen have been identified, of which type I collagen is the most common and constitutes 90% of the protein in the human body [28]. Different types of collagen have a specific form and are found in specific tissues of the body [29]. The structure of type I collagen is based on a triple helix consisting of repeating units of glycine, proline, and hydroxyproline, which provides mechanical strength of tissues and resistance to proteolytic degradation [28,29]. Collagen helices are characterized by interlacing of three polypeptide chains, creating a compact right-handed structure. The central place in the sequence is occupied by glycine, which, due to its small size, allows for tight packing of the chains. The presence of hydroxyproline additionally stabilizes the structure, allowing collagen to maintain exceptional mechanical strength and resistance to enzymatic degradation [30].

Post-translational modifications are essential for the proper structure and function of collagen. Among them, hydroxylation of specific proline and lysine residues is a critical biochemical step that occurs in the endoplasmic reticulum during collagen biosynthesis. This process is catalyzed by prolyl and lysyl hydroxylases and requires cofactors such as iron, α-ketoglutarate, and ascorbic acid [29,31]. Hydroxylation of proline results in the creation of hydroxyproline, which stabilizes the triple helix by enabling the formation of intramolecular hydrogen bonds, thereby enhancing the thermal and mechanical stability of collagen [32]. Hydroxylated lysine residues, in turn, serve as essential sites for glycosylation and subsequent covalent intermolecular cross-linking, which reinforces fibril organization and improves resistance to enzymatic degradation [29]. These biochemical modifications are especially important for the formation of mature and mechanically robust collagen fibrils. The degree of hydroxylation and cross-linking directly influences the structural properties of collagen, including its stiffness, tensile strength, and long-term stability in biological environments. Therefore, proline and lysine hydroxylation plays a critical role not only in the stabilization of individual collagen molecules but also in the assembly and functional performance of collagen-based materials in regenerative and therapeutic applications.

Mechanical properties of collagen are crucial for the functioning of a connective tissue, and closely related to its structure and organization. Collagen type I provides high tensile strength in structures such as tendons, ligaments, and bones, which results from its specific arrangement in ordered fasciculi, i.e., collagen fiber bundles constituting the basic structural units of the tendon. Fasciculi, surrounded by loose connective tissue, allow fibers to slide relative to each other, increasing the elasticity of the tendon during stretching. This organization allows tendons to store mechanical energy and release it effectively, which is crucial in dynamic movements such as running or jumping [33,34]. Type III collagen, due to its elasticity, is found in tissues that require stretchability, such as skin or blood vessels. Although less durable, type III collagen plays an important role in ensuring tissue elasticity. More extensible than type I collagen, type III collagen fibers dominate in the initial stages of wound healing and adaptation of damaged tissues. During regeneration, the synthesis of type III collagen increases, which reduces the mechanical stiffness of tendons and ligaments. Over time, it is replaced by type I collagen, which restores the mechanical strength and functionality of the tissues [35]. The balance between collagen type I and III is necessary to maintain tissue health, and its disruption may lead to pathological changes, such as increased stiffness of muscles or vessels [29].

In recent years, growing attention has been paid not only to the structural roles of collagen types I and III but also to their dynamic remodeling in both regenerative and pathological contexts [36]. These two fibrillar collagens, as major components of the ECM, play essential roles in maintaining tissue integrity, mediating cell signaling, and responding to mechanical and biochemical stimuli during tissue adaptation, repair, and disease progression.

During wound healing and tissue regeneration, type III collagen is initially synthesized to support tissue elasticity and structural reorganization. It forms a more extensible and loosely packed matrix that facilitates cell migration and angiogenesis. As healing progresses, type III collagen is gradually replaced by type I collagen, which restores mechanical strength and durability to the regenerated tissue [29,32]. This transition is tightly regulated by cytokines, growth factors, and post-translational modifications, which stabilize the triple-helical structure and enable enzymatic cross-linking via lysyl oxidase [32]. The coordinated degradation of ECM components, including collagen, is mediated by matrix metalloproteinases (MMPs), such as MMP-1 and MMP-8, which selectively cleave type I and III fibrils and are critical for effective tissue remodeling [29].

Disruption of the balance between collagen synthesis and degradation can lead to pathological fibrosis. In idiopathic pulmonary fibrosis, for example, elevated turnover of collagen types I and III measured by serum biomarkers like C1M and PRO-C3 has been shown to correlate with disease progression and worsening lung function [37]. Similar alterations in the collagen I/III ratio have been observed in cardiac remodeling, where an excess of type I collagen leads to increased myocardial stiffness and impaired diastolic function, while type III is associated with elasticity and compliance [29]. These processes are highly dependent on cell type (e.g., fibroblasts and myofibroblasts), local inflammation, oxidative stress, and mechanical strain. Dysregulated collagen remodeling not only reflects ongoing pathology but may actively contribute to disease progression. Therefore, a deeper understanding of the biochemical and molecular mechanisms governing collagen dynamics is essential for the development of advanced therapeutic strategies in tissue regeneration, wound healing, and fibrosis-related diseases.

### 4.2. Cross-Linking of Collagen

Collagen can undergo the process of cross-linking, which aims to strengthen its structure by creating additional physical and chemical connections between protein chains (Figure 4) [38]. The degree of cross-linking within the collagen matrix affects both its pore size and mechanical stiffness, thereby influencing cellular infiltration. Depending on the method used, different mechanical, chemical, and biological properties of the final material are obtained. The main cross-linking methods are divided into three categories: chemical, enzymatic, and physical [39].

Chemical methods of collagen cross-linking involve the formation of additional bonds in the collagen structure by chemical reactions, which lead to the stabilization of collagen fibers. Chemical cross-linking allows for the modification of mechanical properties, biocompatibility, and stability of collagen materials, which is of key importance in biomaterials and medical applications [40]. Chemical cross-linking, commonly achieved with agents such as glutaraldehyde, results in dense and highly stabilized collagen networks. While this improves the structural strength and resistance to enzymatic degradation, it often leads to an undesired burst release of drugs immediately after implantation. This is due to limited pore size and rapid solvent influx into the tightly packed matrix. Furthermore, the residual toxicity of glutaraldehyde and its calcification-promoting properties limit its biomedical application. Alternative methods include the use of compounds such as diphenylphosphorylazide and 1-ethyl-3-(3-dimethylaminopropyl)carbodiimide, which do not introduce foreign structures into the tissue, but activate carboxyl groups (present in glutamic acid and aspartic acid) to form bonds with amino groups (present in lysine and hydroxylysine) [41] without introducing cytotoxic residues [41,42].

Chemical cross-linking can be also initiated by physical factors. For instance, UV irradiation, γ-radiation, or mechanical compression are able to generate reactive species or induce polymer alignment to create intermolecular bonds. Among them, UV irradiation, usually at 254 nm at the intensity of 0.05–1 J/cm^2^ [43], induces free radical formation and cross-links amino acid side chains. While UV cross-linking improves resistance to degradation, it may cause surface oxidation and mechanical alterations [41]. Another technique is ionizing radiation, which initiates chemical reactions, leading to the formation of a network structure. The typical exposure range is between 10 and 100 Gy [44].

Physical cross-linking of collagen can be achieved by various methods, which involve the use of physical forces to modify the structure of the material. Physical cross-linking takes place when chains are interacting with one another through physical interactions, e.g., chain entanglement [39]. Dehydrothermal (DHT) treatment is commonly employed to stabilize collagen scaffolds without introducing chemical agents. DHT treatment typically involves exposure to dry heat under vacuum, often at 100–120 °C for 12–48 h. For example, a treatment of 105 °C for 24 h under reduced pressure has been shown to enhance mechanical integrity while preserving biocompatibility [39]. However, excessive heat may partially denature the triple helix, affecting functionality. Also mechanical forces, such as stretching or compressing the material, can contribute to the formation of connections between collagen molecules, which results in the reinforcement and stabilization of its structure. These physical methods allow for the control of the mechanical properties of materials by creating a more complex molecular network [45]. Physical cross-linking is solvent-free and allows precise modulation of network density through control of exposure parameters. However, overexposure can excessively densify the matrix and hinder drug diffusion, whereas under-cross-linking risks premature degradation and rapid drug loss. When optimized, physical methods offer tunable control over both mechanical properties and release kinetics [45,46].

Enzymatic cross-linking of collagen is primarily mediated by the enzyme lysyl oxidase (LOX), which catalyzes the oxidative deamination of lysine and hydroxylysine residues to generate reactive aldehyde groups [47]. These aldehydes spontaneously condense with neighboring amino groups to form covalent cross-links that stabilize collagen fibrils, mimicking physiological cross-linking and offering a more biocompatible solution. The efficiency of this process is strongly influenced by several factors. First, the type of collagen plays a role; fibrillar collagens such as type I, which form highly ordered structures, are more favorable substrates for LOX than non-fibrillar types or monomeric forms. Second, post-translational hydroxylation of lysine is critical; hydroxylysine residues are more efficiently oxidized by LOX, and thus a higher degree of hydroxylation enhances cross-linking potential [29]. Third, environmental factors such as pH, oxygen availability, and the presence of cofactors like copper ions (Cu^2+^) are essential for optimal LOX activity [29]. Inadequate copper levels or improper collagen assembly can significantly impair the cross-linking process. The resulting collagen matrix tends to be more porous and loosely structured, supporting more gradual and sustained drug release. In comparison to chemical cross-linking, LOX-mediated systems exhibit smoother release profiles and lower initial burst, especially beneficial for sensitive applications such as brain drug delivery [47,48]. Use of LOX results in networks that are less densely packed and more biocompatible, allowing controlled penetration of glioma cells while minimizing toxicity [47].

Ultimately, the cross-linking strategy should be chosen according to the intended biomedical application, considering mechanical needs, degradation profile, and cell response. Comparative studies (Table 1) indicate that physical methods allow rapid cross-linking but often lack selectivity and may alter collagen conformation. Providing lower cross-linking density and mechanical strength than chemical techniques like glutaraldehyde or EDC/NHS, physical cross-linking methods offer improved cytocompatibility and avoid toxic residues [40,41]. Physical methods are then especially useful for short-term and injectable systems, where low toxicity and preservation of native collagen architecture are desirable. Enzymatic cross-linking using LOX or transglutaminase provides higher physiological relevance and cell compatibility, though the resulting scaffolds are generally less stiff [47]. It also provides the most favorable release characteristics with high biocompatibility, while chemical cross-linking ensures mechanical strength but risks cytotoxicity and burst release, while physical cross-linking offers flexible tuning but requires careful calibration. However, enzymatic methods typically result in slower reaction kinetics and may require longer incubation times or more controlled conditions to achieve comparable mechanical reinforcement. Physical methods, such as UV or γ-irradiation, allow rapid cross-linking but often lack selectivity and may alter collagen conformation.

### 4.3. Collagen as a Drug Carrier

Collagen is a natural biopolymer that, due to its functional groups (carboxylic, amine, hydroxyl), allows drugs to bind in various ways, including hydrogen bonds, as well as electrostatic and hydrophobic interactions. Because of these properties, collagen can be chemically modified, which allows the formation of hybrid materials with various substances, such as anticancer drugs, antibiotics, or hormones. As they can be released in a controlled manner, collagen-based hybrid materials are considered as promising materials in applications related to cancer therapy and other treatments requiring precise dosing [39,40].

The interactions between drug molecules and collagen scaffolds are strongly influenced by the functional groups present on both the biomaterial and the therapeutic agent. These interactions determine drug loading efficiency, release kinetics, and stability. Electrostatic interactions occur when cationic or anionic moieties in the drug interact with oppositely charged amino acid residues in collagen. For example, basic amine groups in drugs such as minocycline or doxorubicin can form ionic bonds with negatively charged carboxyl groups in collagen under physiological conditions [50]. Hydrogen bonding is crucial in the retention of polar drug molecules, particularly those containing hydroxyl or amide functionalities. Temozolomide, due to its imidazotetrazine structure, can form reversible hydrogen bonds with peptide backbones or hydroxyl groups of collagen fibers, promoting sustained retention within the matrix [5]. Hydrophobic interactions also play a role, especially in the case of drugs like irinotecan, which contain aromatic and aliphatic moieties. These interact with non-polar domains of the collagen structure, allowing for their entrapment within the fibrillar network [50].

Apart from the intrinsic chemistry of collagen, the cross-linking method may also affect its interactions with biologically active agents. In chemically modified systems, particularly those utilizing EDC/NHS, covalent conjugation with carboxyl or primary amine-containing drugs is possible. While such binding improves drug stability, it may also limit release kinetics and is, therefore, more appropriate for long-acting or localized applications [40]. The balance between these interaction types is crucial to optimize drug loading and release while maintaining biocompatibility and therapeutic efficacy. Rational selection of drug molecules based on their functional group compatibility with collagen is therefore essential. On the other hand, physical cross-linking techniques, such as DHT treatment and UV irradiation, could decrease the activity of immobilized drugs due to harsh conditions. For instance, DHT treatment typically involves exposure to dry heat under vacuum, often at 100–120 °C for 12–48 h, which could be harmful for some sensitive bioagents. Enzymatic cross-linking using LOX or transglutaminase provides higher physiological relevance and cell compatibility, though the resulting scaffolds are generally less stiff [47].

Exhibiting a fibrillar structure and the ability to enzymatically degrade, collagen ensures stability in the body and precise release of active substances. The organization and density of collagen’s fibrillar network substantially influence both drug loading efficiency and release kinetics. A denser and more ordered fibril arrangement increases the available surface area for drug adsorption through electrostatic interactions, hydrogen bonding, and hydrophobic effects. These interactions are particularly relevant for small-molecule therapeutics and charged bioactive compounds. Highly cross-linked and compact networks—typically achieved using chemical agents such as EDC/NHS or glutaraldehyde—limit interfibrillar spacing and reduce water uptake, resulting in decreased initial burst release and more sustained drug diffusion profiles [40,41]. In contrast, physically cross-linked matrices tend to be less dense and more porous, facilitating faster release but often at the expense of drug retention capacity [39]. Cross-linking density not only affects structural porosity but also modulates mechanical stiffness, which in turn governs molecular diffusivity within the scaffold. Enzymatic cross-linking strategies yield networks with moderate density and improved physiological compatibility, enabling a controlled release without inducing excess rigidity or cytotoxicity [47]. In composite hydrogel systems, such as those based on collagen and hyaluronic acid, the fibrillar architecture of the collagen component plays a central role in governing drug release from embedded liposomal carriers. The spatial organization of fibrils in such matrices can be adjusted to tailor interstitial diffusion, drug partitioning, and overall pharmacokinetic behavior [51]. Optimizing the supramolecular architecture of the collagen network is thus essential for balancing drug entrapment, retention, and controlled release—parameters that are highly dependent on the specific therapeutic application.

### 4.4. Stimuli-Responsive Properties of Collagen Carriers

Since collagen is an amphoteric electrolyte with many acidic or basic side groups, which can dissociate to produce a positive or negative charge in a specific pH range, it is known to exhibit a pH-responsive function [52]. As the tumor site is generally characterized by a lower pH than a healthy tissue, the pH-responsiveness of collagen may be beneficial for the design of targeted drug delivery systems. Indeed, pH-sensitive collagen hydrogels, cross-linked via UV irradiation in the presence of riboflavin, have been shown to reversibly swell and soften under acidic conditions, facilitating controlled drug release [53]. These hydrogels remained stable at physiological pH but become more permeable in acidic environments typical for tumor tissues, thereby enabling more precise spatial drug delivery.

Knowing that many tumors are accompanied with local hyperthermia caused by inflammation and metabolic activity [54], temperature-responsive collagen-based carriers have also been extensively investigated in chemotherapeutics. For instance, thermosresponsive collagen–chitosan composites were found to exhibit potential for brain-targeted therapies, although not precisely against glioblastoma [55]. These injectable systems underwent sol–gel transition at approximately 37 °C, which allowed minimally invasive administration followed by in situ gelation. The resulting scaffold maintained close contact with surrounding tissue and ensured sustained release of the therapeutic payload [55].

## 5. Collagen-Based Drug Carriers for Glioblastoma

The limitations of conventional therapies to completely eliminate GBM cells highlight the importance of targeting the invasive and migratory nature of glioma cells [56]. These cells often infiltrate brain tissue beyond the primary tumor site, making complete surgical removal difficult [57]. Therefore, developing localized and sustained drug delivery systems is a crucial strategy to overcome this challenge [58]. Collagen-based drug delivery platforms provide a promising approach to target the tumor microenvironment and address some of these issues [5]. Being a major structural component of the ECM, collagen plays a vital role in cell signaling, adhesion, and migration. In the case of glioblastoma, the tumor’s interaction with the ECM components is central to its invasive and migratory behavior. Collagen IV, particularly, has been shown to influence tumor cell migration and invasion [5]. Laminin-1, another critical ECM protein, supports cell differentiation, adhesion, and survival, often working synergistically with collagen IV to promote cellular responses that contribute to tumor progression. These components are essential not only for maintaining tissue structure but also for supporting the dynamic interactions between the tumor and its surrounding microenvironment. This makes collagen and other ECM molecules valuable in therapeutic strategies that aim to manipulate or intervene in GBM biology [5].

The GBM microenvironment is characterized by elevated enzymatic activity, particularly involving MMPs, which play a central role in remodeling the ECM. Among them, MMP-1 and MMP-8 are collagenases that degrade fibrillar collagens such as type I and III, while MMP-2 and MMP-9 (gelatinases) target type IV collagen, which is abundant in basement membranes [29]. These enzymes are secreted by both glioma cells and tumor-associated cells, including macrophages and endothelial cells, contributing to aggressive ECM remodeling around the tumor [59]. While this enzymatic landscape facilitates tumor invasiveness and migration [59], it may also negatively affect the structural integrity of collagen-based biomaterials [29,59]. High MMP activity in the tumor core can lead to accelerated degradation of collagen matrices, potentially resulting in uncontrolled or premature drug release and a shortened therapeutic window [29]. On the other hand, this property may be harnessed in the design of so-called “smart” delivery systems that are responsive to local enzymatic activity, enabling targeted and condition-dependent release of therapeutic agents [59]. Understanding the interplay between collagen-based drug delivery platforms and tumor-associated enzymes is essential for the development of more stable, effective, and tumor-selective treatment strategies [29,59].

Collagen-based materials have been shown as effective scaffolds for localized delivery, particularly in GBM therapy, due to their ability to provide sustained release of various agents. In the pioneering work of Shin et al. [60], collagen hydrogel was used as a matrix for the lentiviral vectors immobilized within hydroxylapatite nanoparticles. Formed under relatively mild conditions (incubation at 37 °C for 30 min), acid-soluble rat tail collagen type I served as the primary structure for localized delivery, being able to efficiently maintain the activity of the virus. Depending on the collagen solid content (from 0.05% wt./vol to 0.3% wt./vol), it was possible to modulate the release of lentivirus particles, reaching 13%, 7%, and 5%, respectively, at the first day of release. Since lentivirus was not able to bind to collagen directly, the control of the release was based on the physical properties of the collagen matrix, and precisely its pore size. When implanted subcutaneously into the mice, collagen hydrogel carriers were found to promote long-term and localized expression in vivo for the period up to 4 weeks, showing the ability to maintain therapeutic concentrations at the tumor site for longer durations (Figure 5). This controlled and prolonged release is especially valuable in GBM treatment, where continuous exposure to an anticancer agent is essential to effectively target invasive glioma cells.

In a similar manner, the role of collagen in developing advanced drug delivery systems for cancer treatment—particularly brain tumors—was investigated, with a focus on enhancing its compatibility with drugs and physical resemblance with a tumor microenvironment. It was Jain et al. [61] who highlighted that the invasive nature of GBM requires therapies that not only deliver drugs effectively but also modify the tumor microenvironment to limit tumor progression. Accordingly, a polycaprolactone-based nanofiber system was developed to guide the tumor cells away from the primary tumor site (Figure 6). A new location was called an “extracortial sink” and was made of a collagen-based hydrogel conjugated with cyclopamine—a plant steroid alkaloid that can inhibit the proliferation of tumor cells. When implanted in the vicinity of an intracortical human glioblastoma xenograft, a significant number of cells was observed to migrate to the extracortical collagen site, reducing tumor volume and demonstrating the efficiency of the overall concept. Here, the role of collagen was multimodal. Apart from mimicking the tumor microenvironment, collagen served as an efficient carrier for cyclopamine, since it could be easily tethered with this drug, extending its activity in inducing apoptosis of tumor cells.

The use of collagen-based drug delivery systems in glioblastoma treatment was further validated through the development of a hydrogel system composed of temozolomide-methacrylate (TMZ-MA), poly(ethylene glycol) dimethacrylate (PEGDMA), and N-succinimide methacrylate (Figure 7) [5]. To mimic the ECM of the tumor microenvironment, the surface of the hydrogel was functionalized with collagen IV and laminin-1. This design enabled the hydrogel to function both as a structural scaffold and as a localized drug reservoir. Studies revealed that the incorporation of collagen IV and laminin-1 significantly promoted glioma cell migration toward the hydrogel, since both molecules are known to promote cell adhesion and proliferation. The drug release profile demonstrated that TMZ was gradually released into cerebrospinal fluid (an initial burst release of approximately 20% on day 1, a release of <35% at 3 weeks, and stabilization at <60%) and phosphate-buffered saline (final drug release of 18.48%), simulating physiological conditions and ensuring sustained drug delivery over time. In vivo experiments showed that this system prolonged (up to 30 days) survival and reduced tumor recurrence in mice, providing further support for the effectiveness of collagen-based platforms in glioblastoma therapy.

In another study [50], the authors decided to face the challenges related to the presence of the blood–brain barrier and its limited permeability towards chemotherapeutic agents. Accordingly, the new approach involved the application of local delivery methods as an alternative to the conventional systemic administration of chemotherapeutic agents. The use of collagen type I membranes loaded with irinotecan—a topoisomerase I inhibitor—or minocycline, an antibiotic known for its anti-inflammatory and anti-MMP (matrix metalloproteinase) properties, was explored in a study involving collagen extracted from bovine dermis and cross-linked with glutaraldehyde to form a stable yet biodegradable matrix. The drugs were incorporated at varying concentrations (from 10% to 40% relative to the pure collagen), and the membranes were fabricated by casting and drying the collagen-based mixture. Due to the balance between water swelling and enzymatic degradation, the systems were able to deliver minocycline for 24 h and irinotecan for 2 days. By directly administering therapeutics into the tumor bed, these collagen-based systems were expected to provide higher local drug concentrations with minimal systemic exposure, thus reducing adverse side effects associated with systemic drug delivery.

Although both studies [5,50] represent the use of collagen hydrogels in GBM treatment, their mechanism of action and design differ. The hydrogel described in [5], which is a three-dimensional porous structure, can physically capture migrating glioma cells, offering a “trap-and-kill” mechanism. Its ECM-mimicking properties supported glioma cell migration while simultaneously releasing TMZ to eliminate the infiltrating cells. In contrast, the collagen membranes described in [50] served more as a sustained-release platform, passively modulating the tumor environment without actively recruiting cells. As an alternative, Yue et al. [51] developed an injectable hydrogel-based platform made of collagen and hyaluronic acid for the sustained release of TMZ-loaded liposomes. In contrast to [5,50], in which solid materials were used as drug carriers, the use of in situ gelling hydrogel (able to undergo gelling within 1 min at 37 °C) allowed for the injection of the drug carrier directly in the tumor site, which guaranteed minimal invasiveness of the procedure and perfect filling of the tumor cavity after resection. The use of Type-II collagen and sodium hyaluronate as the matrix components was an efficient way to produce materials able to release between 35% and 80% of TMZ-loaded liposomes over one month. Using 3D tumor-mimic spheroids as an in vitro model, TMZ-loaded liposome/hydrogel composites were shown to exhibit a strong anti-GBM effect and inhibition of the invasiveness of glioma cells. Interestingly, by the proper choice of the hyaluronate-to-collagen weight ratio, it was possible to modulate the expansion of the interstitial space within the tumor spheroids, resulting in the controlled penetration in the spheroids. Overall, this new drug administration strategy was found as a versatile and efficient method to inhibit glioma recurrence and promote glioma apoptosis, acting particularly well as a postoperative adjuvant therapy. The comparison of the performance of collagen-based carriers, used for glioblastoma treatment, in which collagen was applied in different physical forms ranging from solid hydrogels, through solid membranes to liquid gels, has been placed in Table 2.

## 6. Conclusions and Perspectives

Glioblastoma multiforme remains one of the most aggressive and lethal brain tumors, with a median survival time of just 15 months following diagnosis, despite aggressive treatment protocols such as maximal surgical resection, radiotherapy, and chemotherapy. The failure of conventional therapies to completely eliminate tumor cells highlights the importance of targeting the invasive and migratory nature of glioma cells. Therefore, developing localized and sustained drug delivery systems is a crucial strategy to overcome this challenge. Collagen-based drug delivery systems represent a significant advancement in the treatment of GBM. The potential of collagen scaffolds and hydrogels to enable localized, sustained, and targeted delivery of chemotherapeutic agents has been emphasized in multiple studies [5,50,51,60,61]. These therapies could be further optimized for clinical use by integrating collagen-based systems with nanoparticles or stimuli-responsive polymers. Another possibility is to use collagen as a carrier of cesium-131, serving as a surgically targeted radiation therapy device to deliver radiation therapy [62,63,64]. Although not yet used for the treatment of GBM, this approach has been shown efficient in brachytherapy for many non-central nervous system tumors, and lately also for brain tumors. By expressing dual functionality as a seed carrier and a spacer separating the radiation source and a brain tissue, collagen has been shown as safe and efficient as anticancer matrices. Although not yet investigate, collagen carriers could also benefit other anticancer therapies, such as immunotherapy or targeted RNAi technology. It is anticipated that collagen could act as a carrier for a variety of agents used in immunotherapy, including immune checkpoint inhibitors, viruses, or cancer vaccines [65], as preliminarily introduced in [60]. Furthermore, collagen carriers could be potentially applicable in RNAi technology by serving as carriers for small interfering RNA, as already investigated, e.g., for skin regeneration [66].

It is well-known that the tumor microenvironment in glioblastoma is characterized by a reduced extracellular pH (6.5–6.8) and local hyperthermia caused by inflammation and metabolic activity. These parameters could be easily utilized in the design of stimuli-responsive drug delivery systems, enhancing site-specific drug release while preserving systemic biocompatibility. Although stimuli-responsive properties of collagen carriers have not been yet utilized in GBM treatment, their efficiency in targeting other chemotherapeutic sites [53,55] suggests their potential application also in this field. Closely resembling the native ECM, collagen, particularly type IV, serves as a useful scaffold in localized glioblastoma therapy. However, its pro-migratory nature—facilitating cell adhesion and directed migration—can also contribute to tumor cell infiltration rather than containment [5]. This dual functionality introduces a fundamental design conflict: enabling effective recruitment and eradication of infiltrating glioma cells while simultaneously preventing their uncontrolled dissemination into adjacent tissue. One strategy to address this issue involved localization of chemotactic signals within defined scaffold regions, as presented by Zhu et al. [5]. A hydrogel functionalized with collagen IV and laminin-1 was shown to attract tumor cells toward internal scaffold pores, where drug exposure occurred. Confinement of these cues to restricted domains helped reduce off-target migration. Integration of anti-invasive molecules, including MMP inhibitors, represents another strategy. Co-delivery of minocycline with irinotecan in a collagen membrane was reported to stabilize the tumor environment and reduce enzymatically driven matrix degradation and invasion [50].

An exciting opportunity arises when treatment is combined with therapeutic visualization in a theranostics approach, as can be shown on the example of contrast-enhanced collagen carriers. For instance, gadolinium-chelating probes with affinity for collagen type I have demonstrated strong T_1_-weighted MRI signals in fibrotic and pathological tissue, providing localization and real-time monitoring of the carrier system [67]. Similarly, superparamagnetic iron oxide nanoparticles incorporated into collagen scaffolds allow T_2_*-weighted MRI contrast, without compromising scaffold architecture or cytocompatibility. These multifunctional systems support both magnetic targeting and non-invasive imaging and may be adapted for glioblastoma-specific drug delivery [68]. These advancements illustrate the potential of collagen-based carriers to integrate responsive release mechanisms with diagnostic imaging, contributing to the development of more effective and personalized strategies for glioblastoma treatment.

While most studies on collagen-based drug delivery systems for glioblastoma remain in the preclinical domain, early clinical investigations in other therapeutic areas provide encouraging insights into their translational potential. In particular, intracavity and intratumoral delivery systems using collagen matrices have been tested in phase I/II trials for conditions such as osteoarthritis and solid tumors, demonstrating good biocompatibility, prolonged local drug retention, and low systemic toxicity [69,70]. Moreover, prior FDA approval of intracranial, biodegradable polymer wafers (e.g., Gliadel) confirms the feasibility of local drug delivery in the brain, creating a precedent for collagen-based scaffolds [71]. Though specific phase I/II trials on collagen drug carriers for GBM are still lacking, the accumulated clinical data from related systems suggest a promising safety profile and pharmacokinetic behavior, justifying further translational research in this direction.

Collectively, current studies strongly suggest that collagen-based drug delivery platforms could enhance glioblastoma treatment by improving survival rates, reducing recurrence, and minimizing systemic toxicity.

## Figures and Tables

**Figure 1 ijms-26-06513-f001:**
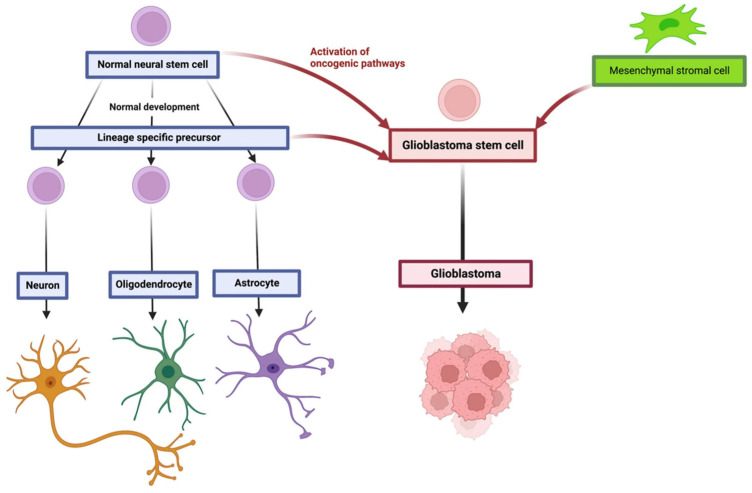
The origin of glioblastoma. Reprinted with permission from [9]. Copyright (2023) Springer Nature.

**Figure 2 ijms-26-06513-f002:**
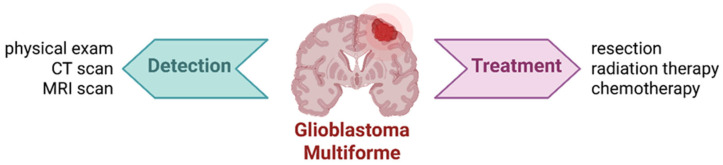
Treatment options for newly diagnosed glioblastoma. Created with BioRender.com.

**Figure 3 ijms-26-06513-f003:**
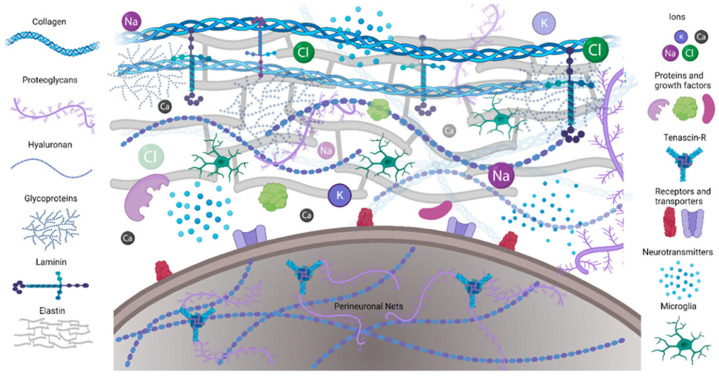
Schematic illustrating the ECM composition. Reprinted with permission from [23]. Copyright (2023) MDPI.

**Figure 4 ijms-26-06513-f004:**
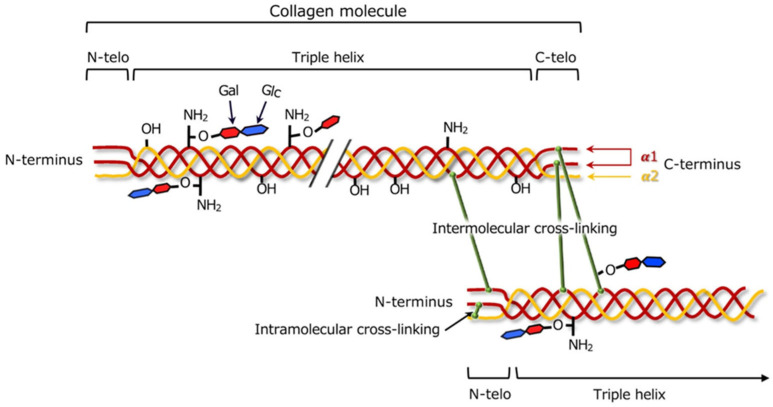
Illustration of type I collagen cross-links formed between two adjacent molecules. Reprinted with permission from [38]. Copyright (2023) Wiley.

**Figure 5 ijms-26-06513-f005:**
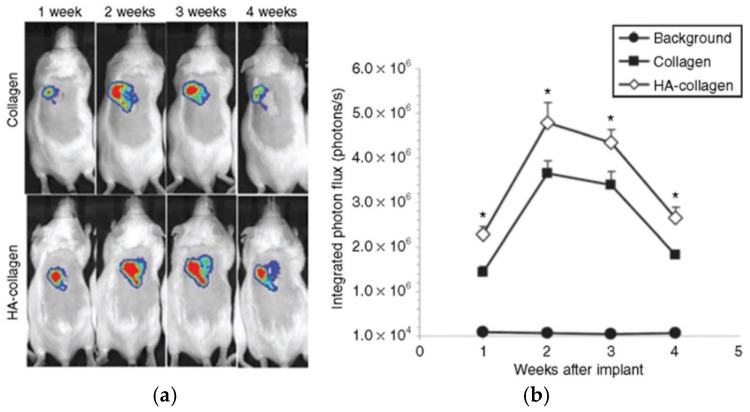
In vivo cell transduction by lentivirus in collagen gels. (**a**) Imaging and quantification of firefly luciferase expression for 4 weeks following subcutaneous implantation of lentivirus or hydroxylapatite (HA)/lentivirus-loaded collagen gels. Lentivirus encoding luciferase (Lenti-luc, 3 × 108 lentivirus particles) was loaded during gelation. (**b**) Integrated light flux (photons/second) as measured using constant-size regions of interest over the implant site (n = 4 for experimental and background data). Collagen gels loaded with lentivirus encoding luciferase (closed squares), HA/lentivirus (open diamonds), and background (closed circles). Values are mean ± SEM. * Statistical significance relative to control group at *p* < 0.05. Reprinted with permission from [60], Copyright (2010) Cell Press.

**Figure 6 ijms-26-06513-f006:**
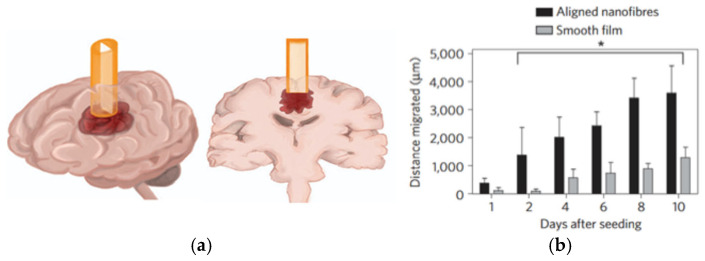
Schematic and image of conduit inserted into a rat brain. (**a**) Representation of the tumor guide containing a nanofiber film inserted into a rat brain. Three-dimensional view (**left**) and coronal view (**right**) of the brain and conduit. (**b**) The graph demonstrates that the distance the tumor cells migrated over ten days was statistically significant on the aligned nanofibers compared with the distance migrated on the smooth film. Mean and standard deviation are shown. Statistical difference was determined using Tukey’s test. A value of * *p* < 0.05 was considered to be statistically significant. Reprinted with permission from [61]. Copyright (2014) Springer Nature.

**Figure 7 ijms-26-06513-f007:**
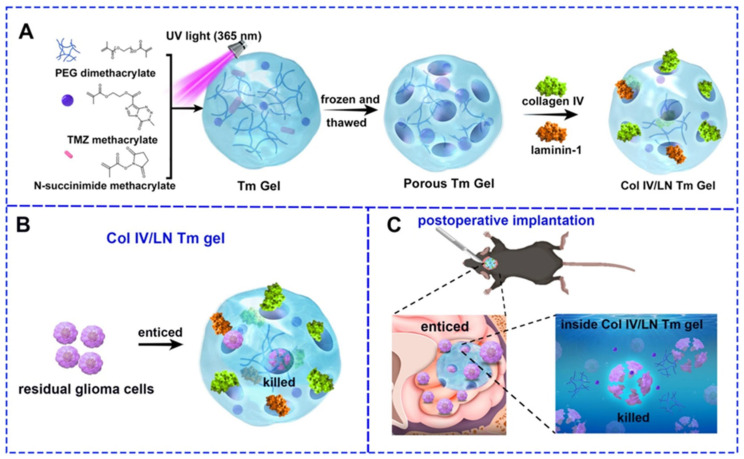
Synthesis of Col IV/LN Tm gel and its effect on inhibiting glioma recurrence. (**A**) PEG dimethacrylate, N-succinimide methacrylate, and TMZ methacrylate formed Tm gel under ultraviolet (UV) light. The Tm gel was repeatedly frozen and thawed for holes. Then, collagen IV and laminin-1 were incubated with Tm gel to form Col IV/LN Tm gel. (**B**) The collagen IV/laminin-1 layer of Col IV/LN Tm gel enticed residual glioma cells to migrate into holes, and then the inner TMZ released to kill recruited glioma cells for inhibiting the recurrence of glioma after surgery. (**C**) Anti-tumor effects of Col IV/LN Tm gel. Residual glioma cells were drawn into the holes by the presence of collagen IV and laminin-1 and subsequently eliminated when the contained TMZ inside the hydrogel was released. Reprinted with permission from [5]. Copyright (2025) Elsevier.

**Table 1 ijms-26-06513-t001:** Comparison of different methods of collagen cross-linking.

Cross-Linking Method	Cross-Linking Agents	Mechanism	Advantages	Limitations	Reference
Chemical	Glutaraldehyde, EDC/NHS	Covalent bonding via amino/acid groups	Strong bonds, tunable stiffness	Residual toxicity (e.g., glutaraldehyde)	[40,41]
Physical	Dehydrothermal treatment, mechanical forces	Hydrogen bonding, chain entanglement	No chemical reagents, low cost	Weak stability, uncontrolled cross-linking	[39,45]
Enzymatic	Lysyl oxidase, transglutaminase	Enzyme-mediated covalent cross-links	Biocompatible, residue-free, ECM-like networks	Sensitive to environment, cost	[47,49]

**Table 2 ijms-26-06513-t002:** Performance comparison of described collagen-based carriers used for glioblastoma treatment.

Collagen Form	Anticancer Agent	Implantation Method	Release Characteristics	Reference
Hydrogel	Lentivirus	Subcutaneous implantation	Stable release over 4 weeks	[60]
	Cyclopamine	Intracortical conduit implant	>10 days sustained release	[61]
	Temozolomide	Gelation in resection cavity	Up to 30 days, 60% total release	[5]
Membrane	Irinotecan, Minocycline	Implantation at tumor site	1–2 days	[50]
Injectable gel	Temozolomide-loaded liposomes	In situ gelling (37 °C)	Controlled release over 1 month	[51]

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
