# Peer review of "Collagen-Based Drug Delivery Agents for Glioblastoma Multiforme Treatment"

_ijms, 2025, doi:10.3390/ijms26136513_

Round 1

Reviewer 1 Report

Comments and Suggestions for Authors
  1. In this manuscript, the authors present an interesting viewpoint on the collagen-based drug delivery agents and related applications, however, a few concerns must be addressed before the next stage of publication.
  2. While experimental examples of collagen carriers are listed (e.g., Shin et al. [42]'s collagen hydrogel, Jain et al. [43]'s nanofiber-collagen system), clinical-stage data (e.g., Phase I/II trials) are lacking. It is recommended to supplementsafety and pharmacokinetic data of collagen carriers from human trials to enhance clinical relevance.
  3. The current focus is on the single function of drug delivery (e.g., sustained release of TMZ [5]). It is suggested to integrateimmunotherapy (e.g., CAR-T [21]) or targeted RNAi technology [18-19], analyzing how collagen carriers could synergistically enhance efficacy.
  4. In Section 4.2 the authors mention crosslinking methods (chemical/enzymatic/physical) but does not compare their impact on drug release profiles. It is recommended to further discuss this issue and clarify whether enzymatic crosslinking (LOX) compared to chemical crosslinking (glutaraldehyde) can reduce the burst release effect.
  5. Carrier morphology varies significantly across studies (hydrogel [5], membrane [44], electrospun fibers [43]), with a lack of performance comparison. It is recommended to add a summary table.
  6. Collagen's promigratory properties(e.g., collagen IV attracting tumor cells mentioned in Section 5) may increase the risk of infiltration. It is recommended to discuss: How to balance the design conflict between trapping tumor cells and preventing dissemination and alternative solutions to crosslinker residue toxicity.
  7. pH/temperature-sensitive collagen carriers (suitable for the tumor microenvironment) are not mentioned and should be discussed.
  8. It is suggested to explore collagen carriers loaded with contrast agents to enable simultaneous treatment and therapeutic visualization.
  9. In the perspective section, recent developments in the multifunctional composite carriers with immunomodulation and imaging functions should be discussed. The importance of large-scale clinical validation should also be mentioned.

Author Response

Reviewer 1

1. In this manuscript, the authors present an interesting viewpoint on the collagen-based drug delivery agents and related applications, however, a few concerns must be addressed before the next stage of publication.

We would like to thank the Reviewer for their positive opinion towards our work. We hope that the revised manuscript will satisfy the Reviewer.

2. While experimental examples of collagen carriers are listed (e.g., Shin et al. [42]'s collagen hydrogel, Jain et al. [43]'s nanofiber-collagen system), clinical-stage data (e.g., Phase I/II trials) are lacking. It is recommended to supplement safety and pharmacokinetic data of collagen carriers from human trials to enhance clinical relevance.

We would like to agree with the Reviewer about the importance of clinical stage data in assessing the suitability of any medical treatment. Unfortunately, up to the best of our knowledge, currently there are no clinical trials on regional chemotherapy of GBM with the use of collagen-based materials. On the other hand, there is plenty of clinical-stage data on collagen carriers used for the treatment of other conditions, e.g. osteoarthritis and specific solid tumors.

We have decided to include them briefly in the text, as a part of the final section of our manuscript (page 17, lines 630-640).

3. The current focus is on the single function of drug delivery (e.g., sustained release of TMZ [5]). It is suggested to integrate immunotherapy (e.g., CAR-T [21]) or targeted RNAi technology [18-19], analyzing how collagen carriers could synergistically enhance efficacy.

Since the application of collagen for GBM treatment is at its early stages, currently collagen has been used solely as a drug delivery agent, without being coupled with neither immunotherapy nor targeted RNAi technology, even though GBM itself could be treated with these approaches. We strongly believe that collagen will become applicable in these fields.

Accordingly, we have decided to add this comment in the “Conclusions and perspectives” section (page 16, line 590-596).

4. In Section 4.2 the authors mention crosslinking methods (chemical/enzymatic/physical) but does not compare their impact on drug release profiles. It is recommended to further discuss this issue and clarify whether enzymatic crosslinking (LOX) compared to chemical crosslinking (glutaraldehyde) can reduce the burst release effect.

According to the suggestion of the Reviewer, the impact on the crosslinking methods on the interactions between immobilized drug and collagen scaffold, governing the release profile, has been added to the text (page 9, lines 349-376). Also a comment on the suitability of enzymatic cross-linking method has been added to the text (page 8, lines 316-318).

5. Carrier morphology varies significantly across studies (hydrogel [5], membrane [44], electrospun fibers [43]), with a lack of performance comparison. It is recommended to add a summary table.

According to the suggestion of the Reviewer, the comparison of the performance of collagen-based carriers, used for glioblastoma treatment, in which collagen was used in different physical forms ranging from solid hydrogels, through solid membranes to liquid gels, has been placed in Table 1 (page 15, lines 570-571).

6. Collagen's promigratory properties(e.g., collagen IV attracting tumor cells mentioned in Section 5) may increase the risk of infiltration. It is recommended to discuss: How to balance the design conflict between trapping tumor cells and preventing dissemination and alternative solutions to crosslinker residue toxicity.

According to the suggestion of the Reviewer, we have added a discussion on the dual functionality of collagen (page 16, lines 604-617) and cross-linker toxicity (page 8, lines 318-319).

7. pH/temperature-sensitive collagen carriers (suitable for the tumor microenvironment) are not mentioned and should be discussed.

According to the Reviewer’s suggestion, we have added a section devoted to stimuli-responsive collagen-based carriers (page 10-11, lines 400-418), and commented on their applicability for glioblastoma treatment in the final section (page 16, lines 597-603).

8. It is suggested to explore collagen carriers loaded with contrast agents to enable simultaneous treatment and therapeutic visualization.

We would like to thank the Reviewer for this suggestion. As theranostics approaches using collagen-based carriers have not been yet used in GBM treatment, we mentioned this exciting opportunity in the final section of our work (page 16, lines 618-629).

9. In the perspective section, recent developments in the multifunctional composite carriers with immunomodulation and imaging functions should be discussed. The importance of large-scale clinical validation should also be mentioned.

According to the suggestion of the Reviewer, the perspective section has been rewritten (page  16-17, lines 590-640).

Reviewer 2 Report

Comments and Suggestions for Authors

The manuscript titled “Collagen-based drug delivery agents for glioblastoma multiforme treatment” presents a study describing the application of collagen as a drug carrier able to provide an extended delivery of chemotherapeutic agents directly to the GBM site, and to provide exciting opportunities for its future applications. However, the manuscript requires major revision before being accepted for publication

As a general suggestion, the manuscript would be significantly strengthened by providing more in-depth explanations of key concepts and supporting the discussion with a broader range of relevant review literature and references. This would enhance both the scientific and the overall clarity of the work.

Some specific suggestions are listed below,

  1. The section offers a clear and informative overview of collagen types I and III, effectively linking their structural characteristics to their mechanical roles in tissues. To enhance clarity and depth, consider minimizing repetitive phrasing and incorporating recent literature on collagen remodeling in pathological or regenerative contexts.
  2. Please rephrase all instances of the phrase 'thanks to' throughout the manuscript. This expression appears frequently and could be replaced with more formal or varied alternatives to enhance the academic tone.
  3. The section effectively highlights the importance of collagen based localized drug delivery in addressing GBM invasiveness. The authors are encouraged to briefly elaborate on the limitations or potential challenges of using collagen scaffolds in clinical translation to provide a balanced perspective.
  4. “The limitations of conventional therapies to completely eliminate GBM cells high light the importance of targeting the invasive and migratory nature of glioma cells. These cells often infiltrate brain tissue beyond the primary tumor site, making complete surgical removal difficult. Therefore, developing localized and sustained drug delivery systems is a crucial strategy to overcome this challenge”, Please provide reference for this line 238 to 241 and rephrase the sentence
  5. The manuscript clearly demonstrates the role of ECM components, particularly collagen, in glioma progression and drug delivery. The authors could consider adding insights into how collagen remodeling by tumor-associated enzymes (e.g., MMPs) might affect the performance of collagen-based delivery systems.
  6. It would be beneficial to include a table summarizing the various collagen-based drug delivery systems that have been investigated for GBM treatment, including those evaluated in preclinical and clinical studies, as reported in the literature.
  7. The manuscript provides a detailed structural and functional overview of collagen types I and III. To enhance the biochemical depth, the authors could briefly discuss the role of post translational modifications such as hydroxylation of proline and lysine in stabilizing the collagen triple helix and enabling intermolecular crosslinking, which are critical for its mechanical properties and resistance to enzymatic degradation.
  8. Consider adding a table for the different type of chemical, physical and enzymatic cross linking that can give a concise information in section 4.2. Also increase the explanation part and literature reference to make the section more detailed
  9. The manuscript mentions the importance of collagen's fibrillar structure in drug delivery applications. Could the authors elaborate on how the organization and density of the fibrillar network influence drug loading capacity and release kinetics in collagen-based delivery systems
  10. Consider expanding Section 4.3 by detailing how the functional groups of drug molecules interact with collagen through various binding mechanisms, and include specific examples of drugs to illustrate these interactions
  11. The description of physical cross-linking methods is useful, but it would benefit from including specific parameters such as radiation dose or mechanical stress applied. Additionally, referencing comparative studies on the efficiency or biocompatibility of these methods would strengthen the section.
  12. The section on enzymatic cross linking provides a good overview of LOX-mediated mechanisms; however, it would be helpful to include more information on how factors such as collagen type, degree of hydroxylation, or environmental conditions influence LOX activity. Citing relevant studies that compare enzymatic cross-linking efficiency with other methods would also add valuable context.

Author Response

Reviewer 2

The manuscript titled “Collagen-based drug delivery agents for glioblastoma multiforme treatment” presents a study describing the application of collagen as a drug carrier able to provide an extended delivery of chemotherapeutic agents directly to the GBM site, and to provide exciting opportunities for its future applications. However, the manuscript requires major revision before being accepted for publication

As a general suggestion, the manuscript would be significantly strengthened by providing more in-depth explanations of key concepts and supporting the discussion with a broader range of relevant review literature and references. This would enhance both the scientific and the overall clarity of the work.

We would like to thank the Reviewer for their opinion towards our work and helpful comments. We hope that the revised manuscript will satisfy the Reviewer.

Some specific suggestions are listed below,

1. The section offers a clear and informative overview of collagen types I and III, effectively linking their structural characteristics to their mechanical roles in tissues. To enhance clarity and depth, consider minimizing repetitive phrasing and incorporating recent literature on collagen remodeling in pathological or regenerative contexts.

According to the suggestion of the Reviewer, the role of collagen remodeling in pathological or regenerative contexts  has been described in the text (page 6, lines 201-229).

2. Please rephrase all instances of the phrase 'thanks to' throughout the manuscript. This expression appears frequently and could be replaced with more formal or varied alternatives to enhance the academic tone.

According to the Reviewer’s suggestion, the overused phrase “thanks to” has been replaced by more relevant expressions.

3. The section effectively highlights the importance of collagen based localized drug delivery in addressing GBM invasiveness. The authors are encouraged to briefly elaborate on the limitations or potential challenges of using collagen scaffolds in clinical translation to provide a balanced perspective.

The final section has been rewritten accordingly (page 16-17, lines 590-640).

4. “The limitations of conventional therapies to completely eliminate GBM cells highlight the importance of targeting the invasive and migratory nature of glioma cells. These cells often infiltrate brain tissue beyond the primary tumor site, making complete surgical removal difficult. Therefore, developing localized and sustained drug delivery systems is a crucial strategy to overcome this challenge”, Please provide reference for this line 238 to 241 and rephrase the sentence

According to the Reviewer’s suggestion, the missing references have been provided.

5. The manuscript clearly demonstrates the role of ECM components, particularly collagen, in glioma progression and drug delivery. The authors could consider adding insights into how collagen remodeling by tumor-associated enzymes (e.g., MMPs) might affect the performance of collagen-based delivery systems.

According to the suggestion of the Reviewer, we have added a section considering the effects of collagen remodeling by tumor-associated enzymes (e.g., MMPs) on the performance of collagen-based delivery systems (page 11, lines 437-452).

6. It would be beneficial to include a table summarizing the various collagen-based drug delivery systems that have been investigated for GBM treatment, including those evaluated in preclinical and clinical studies, as reported in the literature.

According to the suggestion of the Reviewer, the comparison of the performance of collagen-based carriers, used for glioblastoma treatment, in which collagen was used in different physical forms ranging from solid hydrogels, through solid membranes to liquid gels, has been placed in Table 2 (page 15, lines 570-571).

We would like to agree with the Reviewer about the importance of clinical stage data in assessing the suitability of any medical treatment. Unfortunately, up to the best of our knowledge, currently there are no clinical trials on regional chemotherapy of GBM with the use of collagen-based materials. On the other hand, there is plenty of clinical-stage data on collagen carriers used for the treatment of other conditions, e.g. osteoarthritis and specific solid tumors.

We have decided to include them briefly in the text, as a part of the final section of our manuscript (page 17, lines 630-640).

7. The manuscript provides a detailed structural and functional overview of collagen types I and III. To enhance the biochemical depth, the authors could briefly discuss the role of post translational modifications such as hydroxylation of proline and lysine in stabilizing the collagen triple helix and enabling intermolecular crosslinking, which are critical for its mechanical properties and resistance to enzymatic degradation.

According to the suggestion of the Reviewer, the role of post translational modifications has been described in the text (page 5, lines 167-183).

8. Consider adding a table for the different type of chemical, physical and enzymatic cross linking that can give a concise information in section 4.2. Also increase the explanation part and literature reference to make the section more detailed

According to the suggestion of the Reviewer, the section considering cross-linking of the collagen has been extensively revised (page 9, lines 338-339), and now it includes a table comparing different methods of cross linking (page 6-9, lines 230-339).

 9. The manuscript mentions the importance of collagen's fibrillar structure in drug delivery applications. Could the authors elaborate on how the organization and density of the fibrillar network influence drug loading capacity and release kinetics in collagen-based delivery systems

According to the suggestion of the Reviewer, the impact on the collagen's fibrillar structure on drug delivery has been added to the text (page 10, lines 337-399).

10. Consider expanding Section 4.3 by detailing how the functional groups of drug molecules interact with collagen through various binding mechanisms, and include specific examples of drugs to illustrate these interactions

According to the suggestion of the Reviewer, the discussion on the interactions between immobilized drug and collagen scaffold has been added to the text (page 9, lines 349-362).

11. The description of physical cross-linking methods is useful, but it would benefit from including specific parameters such as radiation dose or mechanical stress applied. Additionally, referencing comparative studies on the efficiency or biocompatibility of these methods would strengthen the section.

According to the suggestion of the Reviewer, the section considering cross-linking of the collagen has been extensively revised (page 6-9, lines 230-339).

12. The section on enzymatic cross linking provides a good overview of LOX-mediated mechanisms; however, it would be helpful to include more information on how factors such as collagen type, degree of hydroxylation, or environmental conditions influence LOX activity. Citing relevant studies that compare enzymatic cross-linking efficiency with other methods would also add valuable context.

According to the suggestion of the Reviewer, we have included more information on how different factors influence LOX activity  (page 8, lines 305-313).

Reviewer 3 Report

Comments and Suggestions for Authors

The aim of this paper is to review the use of collagen as carrier for drug delivery and for the treatment of glioblastoma. In the field of biomaterials and more particularly in the field of drug delivery, a huge number of papers are published each year, which bear witness of the importance of the field. By focusing on collagen and the treatment of glioblastoma, the authors limited the number of papers in such a way their description in a review remains manageable. Beyond the biomedical application, the case of glioblastoma is interesting for researchers due to the necessity to take into account the presence of the blood-brain-barrier. This review might be interesting for researchers working in polymer science and more particularly in biomaterials and for those working in medicine. The review is well-written is easy to follow for a broad readership.

My remarks are minor because the manuscript can be easily adapted to take them into account and, accordingly, I recommend the publication of this review with minor corrections.

In line 209, the authors write « Physical cross-linking of collagen can be achieved by various methods, which involve the use of physical forces to modify the structure of the material. One such method is UV radiation, which generates radicals in collagen molecules, leading to the formation of new connections between polymer chains ». Obviously, the authors make a confusion between chemical and physical cross-linking. In macromolecular science, chemical crosslinking takes place when chain are attached to each one another into a giant network through covalent bonds. The formation of the bonds can take place by thermal and photochemical processes.  Physical crosslinking takes place when chains are interracting to each one another through physical interactions but not covalent bonds. In the example given in the review, chemical bonds are created by a photochemical process and it is thus a chemical cross-linking and not a physical one.

In line 231, the authors write “Depending on the needs, collagen carriers can exist in the form of a hydrogel, microspheres, matrices or implants". The sentence is correct but not clear because the authors place on the same level different concepts. Microspheres refers to the morphology of the material. An hydrogel refers to a property, which is the capacity of the material to swell in water without dissolving. You can combine both concepts because hydrogels can be found in films, particles…Finally, implant refers to the application as this is the case for scaffold [for tissue engineering]. My feeling is that sentence could be re-written with an improved structure.

Author Response

Reviewer 3

The aim of this paper is to review the use of collagen as carrier for drug delivery and for the treatment of glioblastoma. In the field of biomaterials and more particularly in the field of drug delivery, a huge number of papers are published each year, which bear witness of the importance of the field. By focusing on collagen and the treatment of glioblastoma, the authors limited the number of papers in such a way their description in a review remains manageable. Beyond the biomedical application, the case of glioblastoma is interesting for researchers due to the necessity to take into account the presence of the blood-brain-barrier. This review might be interesting for researchers working in polymer science and more particularly in biomaterials and for those working in medicine. The review is well-written is easy to follow for a broad readership.

My remarks are minor because the manuscript can be easily adapted to take them into account and, accordingly, I recommend the publication of this review with minor corrections.

We would like to thank the Reviewer for their positive opinion towards our work. We hope that the revised manuscript will satisfy the Reviewer.

In line 209, the authors write « Physical cross-linking of collagen can be achieved by various methods, which involve the use of physical forces to modify the structure of the material. One such method is UV radiation, which generates radicals in collagen molecules, leading to the formation of new connections between polymer chains ». Obviously, the authors make a confusion between chemical and physical cross-linking. In macromolecular science, chemical crosslinking takes place when chain are attached to each one another into a giant network through covalent bonds. The formation of the bonds can take place by thermal and photochemical processes.  Physical crosslinking takes place when chains are interracting to each one another through physical interactions but not covalent bonds. In the example given in the review, chemical bonds are created by a photochemical process and it is thus a chemical cross-linking and not a physical one.

We would like to thank the Reviewer for this clarification. As suggested, we have corrected the description of the physical cross-linking (page 7-8,lines 260-285) 

In line 231, the authors write “Depending on the needs, collagen carriers can exist in the form of a hydrogel, microspheres, matrices or implants". The sentence is correct but not clear because the authors place on the same level different concepts. Microspheres refers to the morphology of the material. An hydrogel refers to a property, which is the capacity of the material to swell in water without dissolving. You can combine both concepts because hydrogels can be found in films, particles…Finally, implant refers to the application as this is the case for scaffold [for tissue engineering]. My feeling is that sentence could be re-written with an improved structure.

According to the suggestion of the Reviewer, the aforementioned sentence has been removed.

Round 2

Reviewer 1 Report

Comments and Suggestions for Authors

The reviewer's concerns has been addressed.